# Microphthalmia-Associated Transcription Factor: A Differentiation Marker in Uveal Melanoma

**DOI:** 10.3390/ijms24108861

**Published:** 2023-05-16

**Authors:** Maria Chiara Gelmi, Robert M. Verdijk, Laurien E. Houtzagers, Pieter A. van der Velden, Wilma G. M. Kroes, Gregorius P. M. Luyten, T. H. Khanh Vu, Martine J. Jager

**Affiliations:** 1Department of Ophthalmology, Leiden University Medical Center, P.O. Box 9600, 2300 RC Leiden, The Netherlands; m.c.gelmi@lumc.nl (M.C.G.); lhoutzagers@gmail.com (L.E.H.); p.a.van_der_velden@lumc.nl (P.A.v.d.V.); g.p.m.luyten@lumc.nl (G.P.M.L.); t.h.k.vu@lumc.nl (T.H.K.V.); 2Department of Pathology, Leiden University Medical Center, 2300 RC Leiden, The Netherlands; r.m.verdijk@lumc.nl; 3Department of Pathology, Section Ophthalmic Pathology, Erasmus MC University Medical Center, 3000 CA Rotterdam, The Netherlands; 4Department of Clinical Genetics, Leiden University Medical Center, 2300 RC Leiden, The Netherlands; w.g.m.kroes@lumc.nl

**Keywords:** uveal melanoma, pigmentation, MITF (Microphthalmia-associated transcription factor), prognosis, chromosome status

## Abstract

Microphthalmia-associated transcription factor (MITF) is an important regulator of melanogenesis and melanocyte development. In cutaneous melanoma, MITF loss has been linked to an increased expression of stem cell markers, a shift in epithelial-to-mesenchymal transition (EMT)-related factors, and increased inflammation. We explored the role of MITF in Uveal Melanoma (UM) using a cohort of 64 patients enucleated at the Leiden University Medical Center. We analysed the relation between MITF expression and clinical, histopathological and genetic features of UM, as well as survival. We performed differential gene expression and gene set enrichment analysis using mRNA microarray data, comparing MITF-low with MITF-high UM. MITF expression was lower in heavily pigmented UM than in lightly pigmented UM (*p* = 0.003), which we confirmed by immunohistochemistry. Furthermore, MITF was significantly lower in UM with monosomy 3/BAP1 loss than in those with disomy 3/no BAP1 loss (*p* < 0.001) and with 8q gain/amplification 8q (*p* = 0.02). Spearman correlation analysis showed that a low MITF expression was associated with an increase in inflammatory markers, hallmark pathways involved in inflammation, and epithelial-mesenchymal transition. Similar to the situation in cutaneous melanoma, we propose that MITF loss in UM is related to de-differentiation to a less favourable EMT profile and inflammation.

## 1. Introduction

Uveal melanoma (UM) is a tumour that arises from uveal melanocytes. It is the most common primary intraocular malignancy in adults and usually involves the choroid (90%) but may also develop in the ciliary body (6%) or the iris (4%) [1,2]. Despite excellent results in local treatment, many patients with UM develop metastases [3,4]. Parameters such as an older age, a large tumour size, the involvement of the ciliary body, and the presence of epithelioid cells are associated with an increased risk of developing metastases [5]. Among the most relevant genetic negative prognostic factors are the loss of chromosome 3, gain of the long arm of chromosome 8, and an inactivating mutation in the *BAP1* (BRCA1 associated protein 1) gene [6,7,8,9,10]. Moreover, it is possible to accurately stratify patients in two metastatic risk categories (Class 1 and Class 2) with a 15-gene expression profile [11,12].

Additionally, an increased tumour pigmentation has been linked to a worse survival in UM patients [13,14,15]. McLean et al. studied the prognosis of small malignant choroidal and ciliary body melanomas and showed that microscopic tumour pigmentation was a negative prognostic factor; they concluded that the prognostic relevance of increased pigmentation was probably due to the higher numbers of epithelioid cells seen in darker tumours [14]. Seddon et al. reported that a heavy microscopic pigmentation in UM was among the five factors that best predicted prognosis, with the most relevant factors being cell type and tumour diameter [15]. These studies did not involve genetic factors. More recently, the Shields group showed that the rate of metastases increased with increasing pigmentation [13]. We recently confirmed the association between a high histologic tumour pigmentation and a lower survival rate in the Leiden enucleation group and showed a link between high levels of pigmentation and loss of chromosome 3/loss of expression of the BAP1 gene [16]. As MITF (microphthalmia-associated transcription factor) plays a major role in regulating pigmentation, we became interested in the function of MITF in UM.

The *MITF* gene is located on the short arm of chromosome 3 and is considered the master regulator of melanogenesis and melanocyte development. Its role in cutaneous melanoma has been studied extensively, as reviewed in Gelmi et al. [17]. In normal melanocytes, MITF stimulates pigmentation, melanocyte development, survival, proliferation, and differentiation [18,19,20,21], while, in cutaneous melanoma, several studies have suggested an association between loss of expression of MITF and the presence of a stem-cell-like phenotype with invasive characteristics [22,23,24]. A study by Matatall in UM cell lines indicated a similar role in UM; *BAP1* loss in UM cell lines led to a decrease in *MITF* and other pigment-related genes and an increase in stem cell markers [25]. Mouriaux reported on primary UM and showed with immunohistochemical (IHC) staining that MITF correlated inversely with the degree of tumour pigmentation but was not related to any other prognostic factor nor with survival in UM patients in a cohort of 57 cases [26]. However, Phelps et al. analysed The Cancer Genome Atlas (TCGA) UM database and reported that UM patients with a low *MITF* mRNA expression had a significantly worse survival than patients with a high *MITF* mRNA expression [27]. Phelps et al. also studied a zebrafish model containing a *GNAQ* and tp53 mutation that had previously been shown to form UM-like tumours in the skin, internal abdomen, and eyes [28]. They showed that *MITF* loss decreased the survival of the zebrafish, suggesting a synergistic effect of *MITF* loss and *GNAQ* mutation in tumour formation and progression in this model [27].

If MITF expression is found to relate to chromosome status in UM, it may also be related to inflammation, as an inflammatory phenotype is typical of UM with a poor prognosis and loss of chromosome 3/BAP1 expression [29,30,31,32,33]. Several inflammatory factors have been reported to influence pigmentation in normal skin, with some cytokines promoting and others inhibiting melanogenesis [34]. Moreover, evidence from the literature shows that normal melanocytes have many other functions in addition to melanin production and that they are involved in the inflammatory process: melanocytes can respond to extracellular signals, can express MHC (Major Histocompatibility Comples) class II molecules, present antigens to the immune system, stimulate phagocytosis, and secrete several cytokines and chemokines [35,36,37,38,39,40,41]. In cutaneous melanoma, evidence points towards an inverse correlation between MITF and inflammation: *MITF* knockdown has shown a correlation with an increased inflammatory phenotype in cutaneous melanoma cell lines, with an inhibitory effect of MITF on the inflammatory response [42].

MITF may not only be involved in inflammation but also in the epithelial–mesenchymal transition (EMT). In cutaneous melanoma, loss of MITF is related to a switch in EMT profile, from a benign and proliferative ZEB2 (zinc finger E-box binding homeobox 2)/SNAI2 (snail family transcriptional repressor 2) positive state to an invasive ZEB1 (zinc finger E-box binding homeobox 1)/TWIST1 (twist family bHLH transcription factor 1) positive state [43,44,45]. In UM, EMT may have a role as well: a high expression of ZEB1, SNAI1 (snail family transcriptional repressor 1), and TWIST1 was shown to increase invasiveness of UM cell lines; ZEB1 expression was higher in high-risk UM cases; and a high TWIST1 expression was correlated with a worse survival [46].

As the length of the follow-up time in the TCGA data, as used by Phelps et al. [27], is limited, and we have the possibility of analysing a patient cohort with mRNA expression and chromosome data with over ten years of follow-up, we set out to determine whether MITF expression was associated with survival in our cohort and whether it was related to prognostic factors, with the tumour’s chromosome status in particular.

Furthermore, we determined the relation between MITF expression and the tumours’ EMT and inflammatory profiles.

## 2. Results

### 2.1. Histopathological Data

In order to determine whether MITF might be relevant to UM progression, we looked at clinical and histopathological data of 64 UM from the Leiden cohort (Table 1). Throughout the manuscript, we will refer to the group of unpigmented and lightly pigmented tumours as “light” and to the group of moderately and heavily pigmented tumours as “dark”. A low *MITF* expression was associated with dark tumour pigmentation (*p* = 0.003), as opposed to light pigmentation (Table 1, Figure 1a). We did not observe an association between *MITF* mRNA expression levels and histological parameters such as cell type, ciliary body involvement, or tumour size (Table 1).

### 2.2. Genetic Associations

We analysed the relation between MITF and several genetic parameters (Table 1): expression of *MITF* was lower in high-risk monosomy 3 (M3) tumours compared to the low-risk disomy 3 (D3) tumours (*p* < 0.001). Similarly, *MITF* was lower in tumours with chromosome 8q gain/amplification compared to UMs with normal 8q status (*p* = 0.02), as well as in tumours with negative BAP1 IHC staining compared to BAP1-positive tumours (*p* = 0.002). Table 1 also shows that *MITF* expression was higher in UM with chromosome 6p gain, which was associated with a better prognosis, albeit with a borderline significance (*p* = 0.049).

As the gene for MITF is located on chromosome 3 and its expression is significantly lower in monosomy 3 than in disomy 3 UM, we revisited the correlation between MITF expression and tumour pigmentation while taking into account the chromosome 3 status. For this purpose, we compared *MITF* expression in four groups of UMs: D3 with light pigmentation, D3 with high pigmentation, M3 with light pigmentation, and M3 with dark pigmentation (Figure 1b). The largest difference in *MITF* expression was seen between D3 and M3 tumours, with the latter having a lower *MITF* expression independently of the degree of pigmentation. The difference in *MITF* expression still existed between D3 tumours with light pigmentation and D3 tumours with a dark pigmentation (*p* = 0.04): dark D3 UM had a significantly lower *MITF* expression than light D3 UM (Figure 1b). However, the number of dark D3 tumours was low (n = 5). Further evidence that chromosome 3 status may not be the only determinant of *MITF* expression is provided in Figure 1c: when considering both chromosome 3 and 8q status, we can see a decreasing trend in *MITF* expression from D3 with normal 8q to D3 with 8q gain to M3. Even though the difference between the first two groups is not statistically significant using the Mann–Whitney U test, it is evident that most of the D3 UMs without 8q gain (9 out of 10) have *MITF* expression > 10.7, while most of the D3 UMs with 8q gain (9 out of 13) have *MITF* expression < 10.7 (Figure 1c).

### 2.3. MITF Expression and Pigmentation

To further examine the relation between MITF and pigmentation, we analysed whether *MITF* gene expression levels were related to the expression levels of several pigment genes. When looking at the mRNA data of our Leiden cohort of 64 UMs, the *MITF* mRNA expression levels showed a significant negative correlation with pigment markers *MC1R* (melanocortin 1 receptor) (*p* = 0.001), *MLANA* (Melan-A) (*p* = 0.04) and *TYRP1* (tyrosinase related protein 1) (*p* = 0.002) and a significant positive correlation with one *RAB27a* probe (*p* = 0.007) (Table 2). We can conclude that the expression of most of the genes involved in melanin synthesis and melanosome development (*MC1R, MLANA,* and *TYRP1*) was inversely related to *MITF* expression.

In order to compare not only mRNA expression but also MITF protein expression with pigmentation levels, we performed IHC staining on sections from 18 UM cases with an anti-MITF monoclonal antibody, which stained both the nucleus and the cytoplasm of UM cells.

We could compare the level of pigmentation with MITF nuclear and cytoplasmic staining. As six tumours had two components with different pigmentation levels, each component was scored for pigmentation and IHC score. The total number of samples was 24: 15 with no/light pigmentation; four were moderately pigmented, and five heavily pigmented. As mentioned earlier, the group including tumours with no and light pigmentation will be referred to as “light pigmentation”.

When looking at the nuclear MITF staining, 14 of the 15 samples with light pigmentations had the highest IHC score (12), while one out of four samples with moderate pigmentations and two of the five heavily pigmented samples had lower scores (light vs. moderate pigmentation *p* = 0.21, no/low vs. heavy pigmentation *p* = 0.053) (Figure 2d). Some cytoplasmic staining was present in all samples, and, in most cases, it was less intense than the nuclear staining. The differences in the MITF between the tumours with different levels of cytoplasmic staining were not significant. When looking at tumours that had two components with different levels of pigmentation, MITF staining was usually higher in the lighter areas than in the darker ones. An example of this kind of tumour is presented in Figure 2. This observation supports the negative correlation between MITF staining and pigmentation, as previously reported by Mouriaux [26].

As a control, we stained for TYRP1. Moderately pigmented and heavily pigmented tumours had a significantly higher TYRP1 score than light tumours (Figure 2e, *p* = 0.03 and *p* = 0.014, respectively). When looking at tumours that had two components with different levels of pigmentation, TYRP1 was usually higher in the darker areas than the lighter ones (Figure 2).

### 2.4. Survival and Pigmentation

We compared UM-related survival between patients with tumours with different degrees of pigmentation. In agreement with prior publications, we found that patients with dark tumours had a significantly worse survival than patients with light tumours (*p* = 0.016) (Figure 3a). The *MITF* expression level was not significantly related to survival when the cohort was split along the median *MITF* expression (Figure 3b) (*p* = 0.30).

However, when looking at the distribution of *MITF* expression, we noticed a steep portion of the curve at the right side, which contained ten cases (15%) with high *MITF* (≥11 Illumina units) (Figure 4a). Therefore, we split the cohort according to this threshold and calculated survival. This “high *MITF*” group had a significantly better survival than the rest of the cases (*p* = 0.03) (Figure 4b). When examining these “high *MITF*” cases more extensively, we noticed that nine of these ten cases had D3.

### 2.5. Inflammation

Several papers on cutaneous melanoma have reported *MITF* silencing to be associated with an inflammatory secretory phenotype [42,47,48]. In UM, an inflammatory phenotype with a predominance of M2 macrophages is associated with a bad prognosis, as opposed to what happens in many other malignancies [29,30,49].

We tested the correlation between the expression of *MITF* and the following inflammatory markers in the Leiden cohort: T cell markers *CD3, CD4*, and *CD8*; Treg marker *FOXP3* (forkhead box P3); macrophage markers *CD68* and *CD163*; and *HLA-A, HLA-B, and HLA-DR* expression (Table 3). Tumours with a low *MITF* expression showed an increase in the numbers of *CD3D* (*p* = 0.048), *CD8A* (probe 3: *p* = 0.01), and *CD68* (two probes, *p* = 0.001 and *p* = 0.004) cells, as well as a higher expression of *HLA-A* (three probes: *p* = 0.003, *p* < 0.001, and *p* = 0.002) and *HLA-B* (*p* = 0.004).

We also performed IHC staining for HLA-DR in the same cases that were stained for MITF and TYRP1. The pattern of HLA-DR positivity suggested that it was mainly expressed in infiltrating cells (Appendix A), while only two samples showed some staining on tumour cells (Appendix A). Interestingly, while the IHC HLA-DR score did not correlate with MITF nucleus IHC score, it showed a positive correlation with the TYRP1 IHC staining (*p* = 0.038) (Appendix A). Moreover, the HLA-DR score for UM with no-low pigmentation was lower than in moderately pigmented and heavily pigmented UM, even though the latter comparison did not reach statistical significance (no-low vs. moderate pigmentation: *p* = 0.027, no-low vs. heavy pigmentation: *p* = 0.053) (Appendix A).

As MITF has been shown to regulate inflammatory cytokines [42,50], we selected the following list of cytokines to test in our cohort: CCL2, CCL3, CCL4, CCL5, CXCL9, CXCL10, CXCL13, IL1B, IL6, IL10, and TNFα. Of the selected cytokines, *CCL2*, *CCL5*, *CXCL10*, and *IL10* had a high enough expression to allow analysis. *MITF* expression showed a negative correlation with *CXCL10* (*p* = 0.033) and with *CCL5* (two probes: *p* = 0.003 and *p* = 0.008) (Appendix A). These findings should be interpreted with caution, and more in-depth analyses are needed since these values may also come from immune cells present in the tumours.

### 2.6. EMT

As previous studies on cutaneous melanoma had shown a possible role of MITF in the switch between EMT states [43,44,45], and Asnaghi used two cohorts, one of which was our dataset, to report that more invasive UM had a higher expression of *TWIST1* [46], we compared *MITF* expression with the presence of EMT and stem cell markers in that same cohort of 64 cases (Table 4). *MITF* showed a significant positive correlation with both probes for *SNAI2* (*p* = 0.001 and *p* = 0.03) and with *ZEB2* (*p* < 0.001) and a negative correlation with *TWIST1* (*p* = 0.04) (Table 4). Our dataset did not include *SNAI1* and *ZEB1*.

### 2.7. Differential Gene Expression Analysis

We carried out a differential expression analysis comparing gene expression in UM with low *MITF* versus those with a high *MITF* expression. The resulting Volcano plot is shown in Figure 5, and the top ten differentially expressed genes are outlined in Appendix A. The most upregulated genes in the low-*MITF* UM were *CXCL16* (C-X-C motif chemokine ligand 16), *RMDN1* (Regulator of microtubule dynamics 1), *HAVCR2* (Hepatitis A virus cellular receptor 2, also known as *TIM3*), *CNIH4* (Cornichon family AMPA receptor auxiliary protein 4), and *HCP5* (HLA complex P5). The most downregulated genes in UM with low MITF were *IRS2* (Insulin receptor substrate 2), *AHCYL2* (Adenosylhomocysteinase like 2), *PPP1R3C* (Protein phosphatase 1 regulatory subunit 3c), *SNHG7* (small nucleolar RNA host gene 7), and *CDV3* (CDV3 homolog).

### 2.8. Gene Set Enrichment Analysis

We used the gene set enrichment analysis to identify differential pathways, comparing UM with low *MITF* to UM with high *MITF*. The results are presented in Figure 6. Among the enriched pathways in UM with low *MITF,* we found several related to immune processes: ALLOGRAFT_REJECTION, INTERFERON_GAMMA_RESPONSE, INFERFERON_ALPHA_RESPONSE, INFLAMMATORY_RESPONSE, COMPLEMENT, IL6_JAK_STAT3_SIGNALLING, and IL2_STAT5_SIGNALING. These results further supported the association between inflammation and a decreased expression of MITF.

Two other pathways that are significantly upregulated in UM with low *MITF* are KRAS_SIGNALING_UP and MTORC1_SIGNALING: these are not pathways that are commonly associated with UM but both play a role in other types of cancer and specifically in metabolic reprogramming [51]. Two further metabolic pathways that are upregulated in low-*MITF* UM are OXIDATIVE_PHOSPHORYLATION and GLYCOLYSIS. This points to a possible metabolic shift in tumours with *MITF* loss. The only pathway that was significantly downregulated in UM with low *MITF* was MITOTIC_SPINDLE.

None of the EMT factors were among the most differentially expressed genes in our differential expression analysis, and the Hallmark EPITHELIAL_MESENCHYMAL_TRANSITION pathway did not show significance (Figure 6).

## 3. Discussion

MITF is a master regulator of melanocyte development, function, and pigmentation and, in cutaneous melanoma, has been linked to a proliferative, non-invasive phenotype. As a mutation in the *MITF* gene is associated with Waardenburg syndrome (which is characterized by a white forelock) [52], and the absence of the mouse homologue mi leads to severe pigmentation defects [19], one would expect non-pigmented tumours to have a low MITF, but Mouriaux [26] previously described a surprising inverse correlation between MITF IHC staining and UM pigmentation. Our study in the Leiden cohort supported the findings by Mouriaux, as we similarly observed an inverse relationship between *MITF* mRNA expression and pigmentation (Table 1 and Table 2, Figure 1 and Figure 2) [26,53,54,55]. These observations may suggest that the non-pigment-related function of MITF may be more relevant than its pigment-related functions in UM.

We now show a relation with the genetic status of the tumour: tumours with monosomy 3, on average, had a lower *MITF* expression than tumours without monosomy 3, as did tumours with negative BAP1 IHC staining compared to UMs with positive BAP1 IHC staining (Table 1). As the MITF gene is present on chromosome 3, it might be that the loss of one copy of chromosome 3 would be responsible for this loss. Both *BAP1* and *MITF* are expressed on the short arm of chromosome 3: *BAP1* on 3p21.1 and *MITF* on 3p13 (Appendix A). This means that both genes may be involved in case of the complete monosomy of chromosome 3 and of the loss of the entire short arm. The frequency of partial chromosome 3 losses varies greatly in different studies, depending on the technique and threshold used, with some authors reporting it to be as low as 4% and others as high as 30%, with diverging opinions on its clinical significance [9,56,57,58,59]. The studies that report the location of partial M3 loss have mostly been carried out with microsatellite analysis: frequently, all the markers on chromosome 3p show alterations (thus potentially involving both *BAP1* and *MITF*), and, when a more limited area is involved, the two regions most frequently altered are 3p24-p26 and 3p12-14.2 (the latter potentially involving *MITF*) [60,61,62,63]. In rare instances, this means that *MITF* might be deleted in cases with disomy 3 and positive BAP1 staining. Moreover, our data on *MITF* expression based on chromosome 3 status and pigmentation show that a negative correlation between *MITF* expression and pigmentation was also present within a set of disomy 3 tumours while chromosome 3 status was more strongly related to *MITF* expression: disomy 3 tumours with low pigmentations had higher *MITF* levels than disomy 3 tumours with high pigmentations, even though we only had five disomy 3 Ums with high pigmentations (Figure 1b). Looking at the gene expression analysis in the Volcano plot, the most upregulated genes in UM with low *MITF* have already been associated with increased malignancy in other cancer types or are related to monosomy 3 in UM (Figure 6, Appendix A). CXCL16 is a chemotactic cytokine; its expression is increased by the activation of Notch and ERK-MAPK (extracellular signal-regulated kinase- mitogen-activated protein kinase) in nasospharyngeal carcinoma and by pro-inflammatory cytokines in prostate cancer, and it has been found to increase proliferation and migration of several types of cancer cells (as reviewed in [64]). RMDN1 enables microtubule binding activity and is expressed on chromosome 8q; we already noticed that UMs with an 8q gain and amplification have a lower MITF expression (Table 1). HAVCR2/TIM3 is an inhibitory receptor of the immunoglobulin superfamily with a specificity for Th1 cells. The expression of HAVCR2/TIM3 and other inhibitory receptors was investigated by scRNAseq in CD8 T cells present in the microenvironment of eight UMs, but that study showed minimal expression of TIM3 [65]. However, TIM3 was among the genes upregulated in hepatic UM metastatic samples compared to a normal liver sample [66]. CNIH4 is a long non-coding RNA that has been found to be downregulated in gastric cancer compared to normal tissue [67], but it has also been reported to facilitate migration in colorectal cancer cells [68]. Lastly, HCP5 is a long non-coding RNA expressed on chromosome 6p; it has been found to promote proliferation and migration in cells of several cancer types (as reviewed in [69]), and it was among the upregulated genes in monosomy 3 UM in a study performed on fine needle biopsy samples [70].

The most downregulated genes offer interesting insights into tumour behaviour as well. We did not find studies about the role of CDV3 in cancer, but it is located on chromosome 3, where MITF is also located. AHCYL2 has been reported to modulate p53-dependent proliferation arrest [71] and to be downregulated in colon carcinoma and lung carcinoma compared to normal lung tissue but not in prostate cancer [72]. Moreover, AHCYL2 has previously been shown to be among the genes downregulated in monosomy 3 UM compared to disomy 3 in a microarray study that included 20 UM samples [73]. SNHG7 is a long non-coding RNA, which has been found to be carcinogenic in some cancer types [74,75,76]. However, a study on the TCGA UM database and UM cell lines reported that SNHG7 was lower in tumours with an epithelioid cell type and in metastatic cases than in UM with spindle cell type and without metastases, and that over-expression inhibited proliferation and induced cell cycle arrest in UM cell lines [77]. These considerations suggested that these two latter genes need to be downregulated in the process of malignant progression in UM. IRS2 and PPP1R3C, are involved in glycogen synthesis, and are among the downregulated genes in monosomy 3 tumours; both had a lower expression in UMs with epithelioid cells in a study performed on UMs using data from the TCGA [33,78]. As shown in Figure 6, the hallmark pathway GLYCOLYSIS was enriched in UMs with low MITFs. This finding and the enrichment in pathways such as OXIDATIVE_PHOSPHORYLATION, KRAS_SIGNALING_UP, and MTORC1_SIGNALING suggested that tumours with low MITF expression may have a different metabolism than tumours with high MITFs. The role of MITF in metabolism has not been extensively studied according to the literature. Some studies have shown an association between MITF and hypoxia and have postulated that hypoxia and nutrient starvation may cause a decrease in MITF and greater invasiveness through HIF1α [23,79,80]. However, the HYPOXIA pathway was not differentially enriched in *MITF*-low vs. *MITF*-high UMs in our cohort. Haq et al. reported that the activation of BRAF/MAPK pathway in cutaneous melanoma cells decreases *MITF*, which, in turn, decreases *PGC1α* (peroxisome proliferator-activated receptor γ coactivator 1), which is a potent activator of oxidative phosphorylation [81]. If a similar mechanism, possibly initiated by BAP1 inactivation, is present in UM as well, it may partially explain the different behaviour of tumours with high and low MITF.

These findings suggest that MITF loss itself may be associated with tumour progression and possibly prognosis.

While a high tumour pigmentation was significantly related to a worse survival (Figure 3a), we did not see a significant difference when we split the two groups along the median of *MITF* expression (Figure 3b). This was similar to the findings of Mouriaux [26]. However, we noticed that a group of patients with very high *MITF* expression had a significantly better survival than all the remaining UM cases in the Leiden cohort (Figure 4). As nine out of these ten cases had disomy 3, this group of UMs might constitute a subgroup of UMs that almost resemble the genetic background of a uveal naevus.

We sought to explain the inverse relationship between MITF and pigmentation and focused on non-pigment-related roles MITF may have in tumour progression: inflammation and EMT. We have previously shown that an inflammatory phenotype in UM is a bad prognostic sign and is associated with monosomy 3 [30,49,82]. A relationship between MITF and inflammation is known to exist in cutaneous melanoma: Arts et al. showed that IL1β decreased *MITF-M* expression, partly through an upregulation of miR-155 in human cutaneous melanoma cell lines, and that samples with high *MITF* and *TYR* (tyrosinase) expressed low *IL1β* in a mouse model and vice versa [47]. Moreover, the modulation of MITF expression in cutaneous melanoma cells has been shown to cause changes in the inflammatory response [42,50]. As far as we know, the relation between MITF and inflammation has not been explored in UM yet. Intriguingly, Souri recently showed that miR-155 expression correlated with the presence of infiltrating CD4 and CD8 T lymphocytes and CD68 macrophages and with an increased expression of HLA-A and HLA-B in UM, all of which indicated the presence of the UM inflammatory phenotype [82,83]. Our analyses demonstrate that there is an inverse relation between *MITF* expression and inflammation in UM as well: tumours with low *MITF* had significantly higher levels of markers of cytotoxic T cells (*CD3, CD8*), macrophages (*CD68*), and HLA Class I molecules, which are markers of the inflammatory phenotype (Table 3). *HLA-DR* did not correlate with *MITF* at mRNA or protein level, but the HLA-DR IHC score was positively correlated with the TYRP1 IHC score and was lower in tumours with no-low pigmentation compared to moderately and heavily pigmented tumours (Appendix A). Expression of HLA-DR and HLA class II in general has been reported on UM tumour cells in the literature [31,84,85], but we observed staining of tumour cells only in small areas in two cases. Therefore, we can assume our HLA-DR IHC score to be representative of immune cell infiltration. *MITF* expression also showed a negative correlation with expression of *CCL5* and *CXCL10*. mRNA expression of both these chemokines was correlated to the T cell fraction in a previous study on the same cohort of 64 cases, which also showed that CXCL10 IHC staining was strong in cases with high T cell count and was observed exclusively in CD163-expressing macrophages [86]. Moreover, gene set enrichment analysis confirmed the upregulation of immune-related pathways in UM with low *MITF* compared to UM with high *MITF* (Figure 6). As these observations are based on in silico correlations, functional studies are needed to confirm if modulation of MITF expression directly leads to a change in the expression of inflammatory markers. A further consideration is that we found the melanocortin receptor *MC1R* among the pigment-related genes with an inverse correlation with *MITF* expressions tested in Table 2. Interestingly, the expression of MC1R has been detected in immune cells as well [87,88,89]. As our mRNA data were measured with bulk RNA microarray, one may hypothesise that the *MC1R* mRNA expression we measured came partly from immune cells instead of melanoma cells. In UM, the presence of T cells and macrophages is known to be associated with poor survival and with loss of chromosome 3 [30,82]. We do not know whether the association between MITF and an increased inflammation is under the influence of MITF or is caused by other genes located on chromosome 3, such as BAP1.

MITF has also been associated with the process of EMT in cutaneous melanoma [44,90]. Caramel et al. and Denecker et al. proposed a model in which ZEB1 and TWIST1 had stem cell and tumour-inducing and pro-invasive properties, while ZEB2 and SNAI2 acted as tumour suppressors and induced differentiation (and MITF expression) [43,44]. Interestingly, the findings from Asnaghi et al. corroborated this model [46]. Our analysis in a series of UM with a long follow-up further supported this link, and, in particular, we reported a statistically significant positive correlation between *MITF* expression and the expression of *SNAI2* and *ZEB2* and a negative correlation with *TWIST1* (Table 4). Caramel et al. stated that the BRAF mutation caused this EMT-TF (transcription factor) rearrangement, but, in UM, a BRAF mutation is very rare. However, in the literature, there is evidence of a link between BAP1 and EMT. One study on cervical cancer cell lines reported that BAP1 knockdown induced increases in N-cadherin and vimentin and a decrease in E-cadherin (which are hallmarks of EMT) and increased cell migration [91]. We postulate that BAP1 inactivation specifically, which is common in UM and carries a bad prognosis, may be involved in the EMT-TF switch in UM and may involve MITF: we noticed a positive correlation between expression of *BAP1* and *SNAI2* and a negative correlation between *BAP1* and *TWIST1*. We saw a correlation between loss of BAP1 IHC and loss of *MITF* mRNA expression and lent support to the theory by Matatall that a *BAP1* mutation in melanocytes induced regression to a stem cell and invasive cell state, as previously developed on the basis of cell lines [25].

In summary, we hypothesised that UMs with low MITF had more invasive phenotypes and less favourable EMT profiles (high TWIST1, SNAI1) than UMs with high MITF (which have high ZEB2 and SNAI2), and that this EMT shift may be initiated by BAP1 inactivation. As stated above, these in silico results should be confirmed by functional studies, possibly through modulation of MITF expression. Further studies on the role of EMT in UM and on its association with MITF may give more insight in the processes that lead to tumour invasion and may help find new therapeutic strategies.

A further point of discussion involves our IHC analyses. Most of our samples showed not only nuclear MITF staining but also cytoplasmic staining. Mouriaux also reported that some of their UM samples showed cytoplasmic MITF staining and attributed this to the presence of MITF-A rather than MITF-M [26]. However, other authors suggested that cytoplasmic MITF staining might not be specific since it was observed in some clear cell sarcomas (which show evidence of melanocytic differentiation) but also breast cancer samples that were non-melanocytic [68,69,70].

As we have shown in our analyses, tumours with high MITF and low MITF are genetically different, and *MITF* expression has an inverse correlation with genetic negative prognostic factors such as monosomy 3, BAP1 loss, and chromosome 8q gain. A low *MITF* expression was also associated with an inflammatory phenotype and a switch in the EMT profile. Hence, we can hypothesise that MITF has a role in UM behaviour, although not directly related to patient survival, and the modulation of its expression in UM cells may be part of the process of tumour progression and metastasis formation.

## 4. Materials and Methods

A retrospective analysis of a database containing data from 64 UM cases enucleated between 1999 and 2008 at the Leiden University Medical Center (LUMC), Leiden, The Netherlands. Clinical information was gathered from patient files, and survival information was gathered from the Integral Cancer Center West patient records and was updated in 2021. Of the 64 patients, 37 died with UM metastases, 17 were alive at the time of the analyses, four died of causes unrelated to UM, and six died of unknown causes.

For each case, part of the tumour tissue was snap frozen with 2-methyl butane and used for DNA and mRNA isolation. The remainder of the tissue was fixed in 4% neutral-buffered formalin for 48 h and embedded in paraffin for histological analysis.

Our analysis focused on gender, age at enucleation, tumour pigmentation, eye colour, largest basal diameter (LBD), thickness, mitotic count, cell type, ciliary body involvement, scleral ingrowth, TNM-AJCC (Tumour-node-metastasis-American Joint Committee on Cancer) stage, chromosome 3 status, chromosome 8q status, chromosome 6 status, and BAP1 status. Tumour pigmentation was assessed macroscopically after enucleation and scored on a 4-point scale (unpigmented to heavily pigmented). Subsequently, if appropriate for the analysis, unpigmented and lightly pigmented UMs were grouped in the “light UM” group, and moderately and heavily pigmented UMs were grouped in the “dark UM” group. LBD, thickness, mitotic count, cell type, ciliary body involvement, and scleral ingrowth were scored histologically.

Chromosome status was determined through single-nucleotide polymorphism analysis with the Affymetrix 250K_NSP-chip and Affymetrix Cytoscan HD chip (Affymetrix, Santa Clara, CA, USA), and chromosome 8 copy number was obtained by droplet digital polymerase chain reaction. Chromosome 8q status was classified as follows: normal 1.9–2.1 copies; gain 2.2–3.1 copies; amplification >3.1 copies [92]. Expression of BAP1 was determined by immunohistology [93,94].

RNA isolation was performed with the RNeasy mini kit (Qiagen, Venlo, The Netherlands), and gene expression analysis was performed with the Illumina HT-12 v4 chip (Illumina, San Diego, CA, USA). For inflammatory markers, we used probes that had previously been validated through immunohistochemical studies [32,95,96]. A list of all the probes used can be found in the Appendix A.

In order to carry out the survival analysis and differential expression analyses, as the distribution of *MITF* expression did not show any clear inflection point, *MITF* expression was classified as low or high, splitting the data along the median; hence, considering the 32 cases with lower *MITF* as “low” and the 32 cases with higher *MITF* as “high”. As there was an inflection point at 11 Illumina units (Figure 4), we used this as a cut-off point to separate cases with very high expression.

### 4.1. Samples for Immunohistochemical (IHC) Staining

Slides for immunohistochemical staining were cut from 18 FFPE tissue samples selected from the 64 UM Leiden cohort. In total, 20 samples were initially selected for IHC staining, 10 with high and 10 with low *MITF* mRNA expression. As two samples had to be excluded because of lack or scarcity of tumour material in the pathological slide, 18 slides were stained.

### 4.2. IHC Staining and Scoring

Immunohistochemistry was performed with an automated, validated, and accredited staining system (Ventana Benchmark ULTRA, Ventana Medical Systems, Tucsen, AZ, USA) using an ultraview universal alkaline phosphatase red detection Kit (#760-501). In brief, following deparaffinization and heat-induced antigen retrieval, the tissue samples were incubated according to their optimized time with the antibody of interest (MITF: anti-mouse, 0.4 µg/mL, Ventana, C4/D5, TYRP1: anti-rabbit, 1/4000, Abcam, Cambridge, UK, EPR21960). Incubation was followed by hematoxylin II counter stain for 12 min and then a blue colouring reagent for 8 min according to the manufacturer’s instructions (Ventana).

The scoring was carried out considering both intensity of staining on a scale from 0 to 3 and percentage of stained cells on a scale from 1 to 4 (1 =< 10%, 2 = 10–50%, 3 = 50–80%, 4 => 80%). The intensity and the cell percentage scores were multiplied, giving a maximum score of 12 for each sample. The level of microscopic pigmentation was scored on a scale from 1 to 4, with 1 being unpigmented and 4 heavily pigmented. In the tumours that had two components with different pigmentation, each component was scored separately. As there were six tumours with a double component, a total of 24 scores were analysed.

### 4.3. Statistical Analyses

Statistical analysis was performed with SPSS, version 25 (IBM Corp., Armonk, NY, USA). The graphs were computed with Graphpad Prism 8 (GraphPad Software, La Jolla, CA, USA). In the first phase of our study, MITF expression was used as a continuous variable, and it was correlated with clinical, histopathological, and genetic features using Mann–Whitney U test for dichotomous variables and Kruskal–Wallis test for variables with more than two groups. As the variables used for these analyses are strongly related to each other, we deemed a multivariable analysis not to be ideal in this scenario. The correlation between gene expression levels was calculated through Spearman’s rho correlation coefficient. Survival was calculated through Kaplan–Meier and log-rank test, and cases who died of another or unknown cause were censored. In our analyses, a *p* value < 0.05 was recorded as significant. The differential expression analysis and gene set enrichment analysis were carried out in the statistical software R. The probe with the highest mean expression was selected for each gene in the microarray. The package limma was used for differential expression analysis, and the significance threshold for the volcano plot included an adjusted *p* value < 0.05 and log FC > 0.6 or <−0.6. The gene set enrichment analysis (GSEA) was performed with the R package fgsea, and significance was established as a *p* value ≤ 0.01.

### 4.4. Study Approval

The study was approved by the Scientific Committee of the Ophthalmology Department of the Leiden University Medical Center (project number 29.1). Tumour material was made available for research according to the Dutch FEDERA regulations of left-over material of pathological specimens. The research adhered to Dutch law and the tenets of the Declaration of Helsinki (World Medical Association of Declaration 2013; ethical principles for medical research involving human subjects).

### 4.5. Limitations

One limitation of this study was the sample size, and the number of cases with unpigmented and highly pigmented tumours was small. Moreover, our analysis focused on enucleated tumours, which are usually larger than UM treated conservatively. Therefore, our results may not be completely reproducible in a cohort with smaller tumours. One further limitation was the retrospective nature of this analysis, which did not allow us to actively select the type of tests to perform or the patient populations.

## 5. Conclusions

We conclude that a low MITF expression profile characterises a subgroup of UM with high pigmentation and inflammation. Furthermore, these tumours have an unfavourable EMT profile and are associated with monosomy of chromosome 3/loss of BAP1 expression. As the relation between low MITF expression and increased pigmentation is also seen in disomy 3 tumours, it can be assumed that there is not only a dose effect but that MITF itself is responsible for these associations.

## Figures and Tables

**Figure 1 ijms-24-08861-f001:**
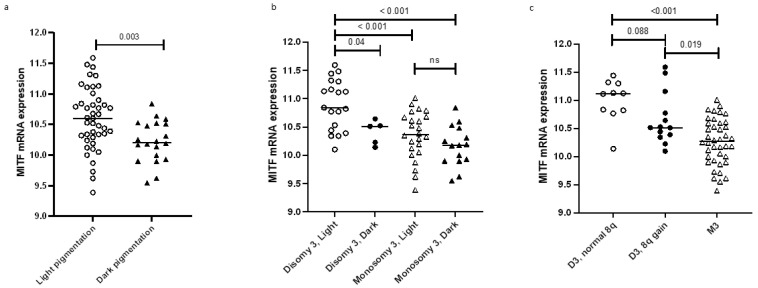
MITF mRNA expression is negatively correlated with macroscopic tumour pigmentation in a set of 64 enucleated UM. Furthermore, MITF expression is significantly lower in monosomy 3 than in disomy 3 UM. (**a**): MITF mRNA expression as determined in an Illumina microarray in 64 cases from the Leiden cohort, divided into tumours with macroscopically light or dark pigmentations (Mann–Whitney U test). (**b**): Distribution of MITF mRNA expression in UMs according to chromosome 3 status plus pigmentation levels in the Leiden 64 UM case cohort, Mann–Whitney U test. (**c**): Distribution of MITF mRNA expression in UMs according to chromosome 3 and 8q status in the Leiden 64 UM case cohort, Mann–Whitney U test.

**Figure 2 ijms-24-08861-f002:**
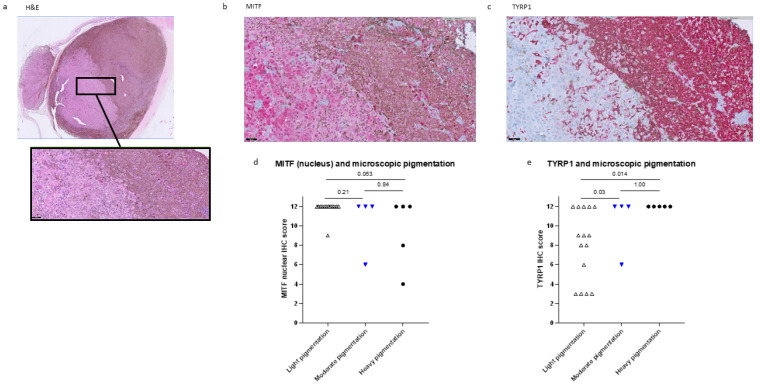
Less MITF IHC staining is seen in heavily pigmented UM. (**a**): A UM case with two components with different pigmentation levels (haematoxylin and eosin) shows higher MITF staining in the light area than in the dark area of the tumour (**b**) and an opposite pattern in TYRP1 staining (**c**). (**d**,**e**): UM with more pigmentation show a lower IHC nuclear MITF score (**d**) and a higher TYRP1 score (**e**). Magnification: 1× is (**a**), 20× in (**a**) (detail, in black box), (**b**,**c**). H&E = Haematoxylin and eosin.

**Figure 3 ijms-24-08861-f003:**
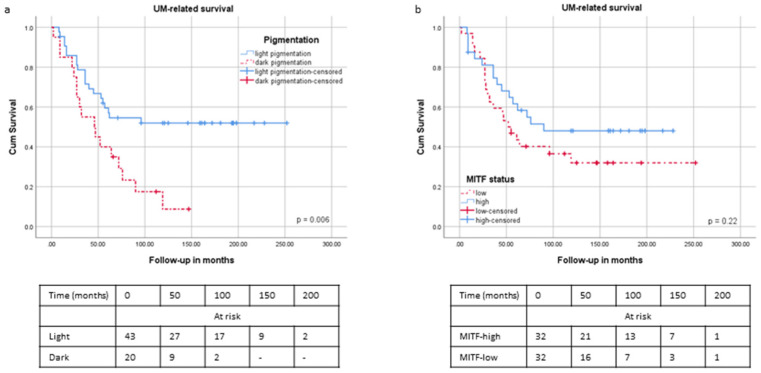
Tumours with moderate/high pigmentation are associated with a worse survival than tumours with no/low pigmentation. (Kaplan–Meier curves and log rank test). (**a**): UM-related survival according to the level of tumour pigmentation (*p* = 0.016); (**b**): UM-related survival according to the level of MITF mRNA expression (*p* = 0.30 with two MITF groups, separated according to the median expression level).

**Figure 4 ijms-24-08861-f004:**
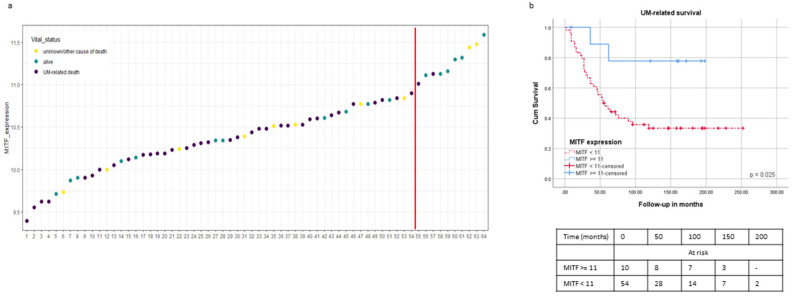
Tumours with very high MITF mRNA expression have excellent prognosis. (**a**): Distribution of MITF mRNA expression across 64 UM samples in the Leiden cohort; the red line indicates the cut-off used in the survival analysis. (**b**): Kaplan–Meier curves showing the UM-related survival in two MITF groups, with expression <11 vs. ≥11 (*p* = 0.031).

**Figure 5 ijms-24-08861-f005:**
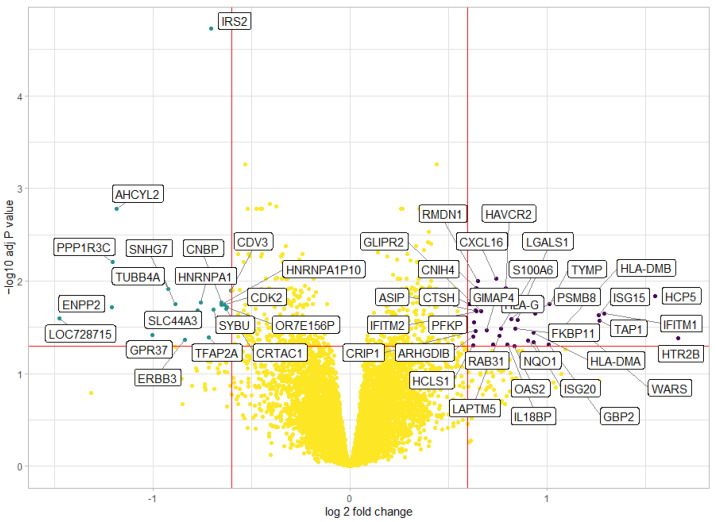
Volcano plot showing genes that are differentially expressed between MITF-low and MITF-high UM in 64 UM. MITF mRNA expression was determined in an Illumina microarray, and the median expression level was used to define the MITF-low and MITF-high groups.

**Figure 6 ijms-24-08861-f006:**
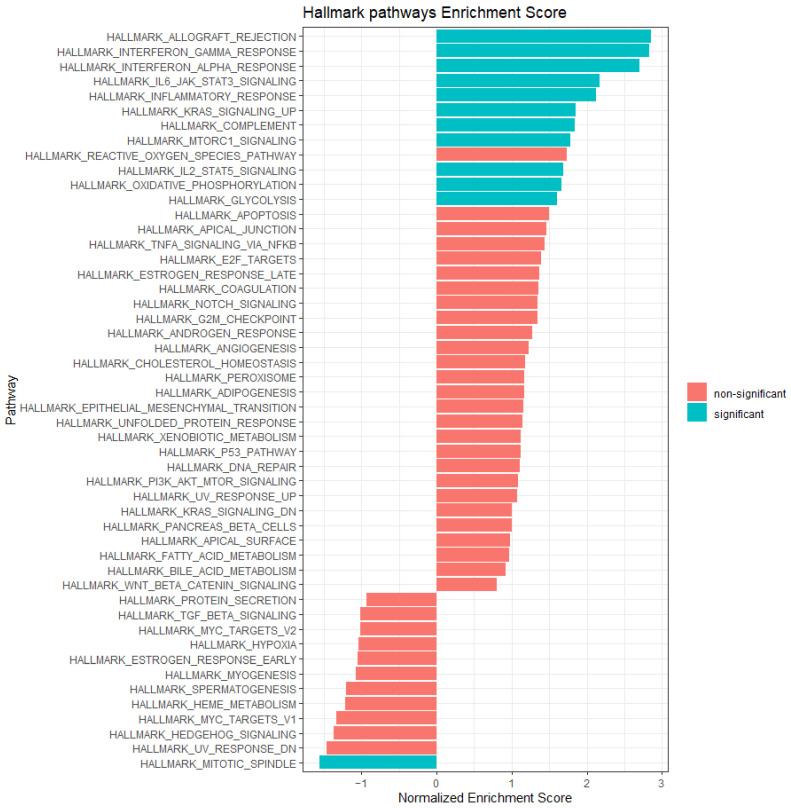
Gene set enrichment analysis comparing UM with low MITF mRNA expression and UM with high MITF mRNA expression in the Leiden cohort (64 cases, separated at the median). Significance threshold: *p* < 0.01.

**Table 1 ijms-24-08861-t001:** Distribution of MITF mRNA expression in UM with different clinicopathological features in 64 UM from the Leiden cohort. MITF mRNA expression as determined in an Illumina microarray. Significant *p* values are shown in bold.

		Median (Min-Max)	
Feature	Nr	MITF Expression	*p* Value
**Gender**			
Male	33	10.44 (9.4–11.3)	0.60 ^a^
Female	31	10.48 (9.6–11.6)	
**Iris colour ^e^**			
Light	29	10.52 (9.6–11.4)	0.28 ^a^
Dark	8	10.25 (10.0–11.1)	
**Tumour pigmentation ^f^**			
Light	43	10.59 (9.4–12.0)	**0.003 ^a^**
Dark	20	10.21 (9.6–10.8)	
**Cell type**			
Spindle cell	22	10.50 (9.6–11.5)	0.76 ^a^
Epithelioid-mixed cell	42	10.41 (9.4–11.6)	
**Ciliary body involvement**			
No	39	10.48 (9.4–11.6)	0.64 ^a^
Yes	25	10.38 (9.6–11.1)	
**TNM stage**			
I-IIB	36	10.50 (9.6–11.6)	0.28 ^a^
IIIA-IIIC	26	10.35 (9.4–11.1)	
**Chromosome 3 status**			
Disomy	24	10.77 (10.1–11.6)	**<0.001 ^a^**
Monosomy	40	10.27 (9.4–11.0)	
**Chromosome 8q status**			
Normal	13	10.84 (9.7–11.4)	**0.02 ^b^**
Gain	23	10.48 (9.6–11.6)	
Amplification	27	10.32 (9.4–11.0)	
**Chromosome 6p status**			
Normal	43	10.35 (9.6–11.6)	**0.049 ^a^**
Gain	21	10.53 (9.4–11.5)	
**BAP1 expression (IHC) ^e^**			
BAP1 positive	25	10.53 (9.6–11.4)	**0.002 ^a^**
BAP1 negative	31	10.35 (9.4–11.5)	
**Age at enucleation**		−0.063 ^c^	0.62 ^d^
**Largest Basal Diameter**		−0.168 ^c^	0.18 ^d^
**Thickness**		−0.122 ^c^	0.34 ^d^

^a^: Mann–Whitney U test; ^b^: Kruskal–Wallis test; ^c^: Spearman’s correlation coefficient; ^d^: Spearman’s correlation; ^e^: Iris colour and BAP1 IHC were not available for all cases; ^f^: light pigmentation includes unpigmented and lightly pigmented UM, dark pigmentation includes moderately and heavily pigmented UM. IHC = Immunohistochemistry. TNM = Tumour Node Metastasis.

**Table 2 ijms-24-08861-t002:** Correlations between mRNA expression of MITF and pigmentation genes in 64 UM patients from the Leiden cohort (microarray). Spearman’s correlation. MITF expression (Mean, ± SD) = 10.46 (±0.5). mRNA expression was determined in an Illumina microarray. Significant *p* values are shown in bold. MC1R = melanocortin 1 receptor; MLANA = Melan-A; PMEL = premelanosome protein; TYR = tyrosinase; TYRP1 = tyrosinase related protein 1; OA1 = ocular albinism type 1.

	Mean (±SD)	Correlation Coeff.	Sig (2-Tailed)
MC1R	7.95 (±0.6)	−0.406	**0.001**
MLANA	13.73 (±0.4)	−0.259	**0.038**
PMEL	14.38 (±0.4)	−0.123	0.33
TYR	12.08 (±0.6)	0.182	0.15
TYRP1	13.66 (±0.9)	−0.379	**0.002**
RAB 27a pr1	8.56 (±0.5)	−0.116	0.36
RAB 27a pr3	7.95 (±0.5)	0.332	**0.007**
OA1	9.84 (±0.6)	−0.106	0.41

**Table 3 ijms-24-08861-t003:** Correlations between mRNA expression of MITF and inflammation genes in 64 UM patients from the Leiden cohort (microarray). Spearman’s correlation. MITF expression (Mean, ± SD) = 10.46 (±0.5). mRNA expression was determined in an Illumina microarray. Significant *p* values are shown in bold. FOXP3 = forkhead box P3.

	Mean (±SD)	Correlation Coeff.	Sig (2-Tailed)
CD3D	7.13 (±1.1)	−0.249	**0.048**
CD3E	6.51 (±0.3)	−0.203	0.11
CD4	6.66 (±0.3)	−0.237	0.06
CD8A pr1	7.19 (±1.1)	−0.220	0.08
CD8A pr3	7.26 (±1.3)	−0.315	**0.011**
CD68 pr1	10.84 (±0.9)	−0.404	**0.001**
CD68 pr2	9.45 (±0.9)	−0.359	**0.004**
CD163 pr3	6.62 (±0.3)	−0.091	0.48
FOXP3	6.50 (±0.1)	−0.057	0.65
HLA-A pr1	11.36 (±1.0)	−0.363	**0.003**
HLA-A pr2	13.84 (±0.8)	−0.429	**<0.001**
HLA-A pr4	10.71 (±1.4)	−0.379	**0.002**
HLA-B	11.33 (±1.7)	−0.358	**0.004**
HLA-DRA pr1	10.49 (±1.4)	−0.153	0.23
HLA-DRA pr2	11.34 (±1.5)	−0.228	0.07

**Table 4 ijms-24-08861-t004:** Correlations between mRNA expression of MITF and EMT-related genes in 64 UM patients from the Leiden cohort (microarray). Spearman’s correlation. MITF expression (Mean, ± SD) = 10.46 (±0.5). mRNA expression was determined in an Illumina microarray. Significant *p* values are shown in bold. ZEB2 = zinc finger E-box binding homeobox 2; SNAI2 = snail family transcriptional repressor 2; ZEB1 = zinc finger E-box binding homeobox 1; TWIST1 = twist family bHLH transcription factor 1; SNAI1 = snail family transcriptional repressor 1; STAT3 = signal transducer and activator of transcription 3.

	Mean (±SD)	Correlation Coeff.	Sig (2-Tailed)
SNAI2 pr1	8.17 (±0.6)	0.413	**0.001**
SNAI2 pr2	9.49 (±0.7)	0.278	**0.026**
ZEB2	7.67 (±0.3)	0.501	**<0.001**
TWIST1	6.60 (±0.2)	−0.263	**0.036**
STAT 3 pr1	8.02 (±0.4)	−0.146	0.25
STAT3 pr2	9.05 (±0.4)	−0.145	0.25

## Data Availability

Data available on request due to restrictions e.g., privacy or ethical.

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
