# Peer review of "Microphthalmia-Associated Transcription Factor: A Differentiation Marker in Uveal Melanoma"

_ijms, 2023, doi:10.3390/ijms24108861_

Round 1

Reviewer 1 Report (New Reviewer)

 The manuscript entitled “Microphthalmia-associated transcription factor: a differentiation marker in Uveal Melanoma” describes an inverse correlation between MITF IHC staining and pigmentation of uveal melanoma, which associated with monosomy of chromosome 3/ loss of BAP1 expression. MITF is a well-known and well-studied regulator of cutaneous melanoma, and only a few studies address it in uveal melanoma. The aim of this study is clear, the results are well written, however, there are many limitations of the study which are stated clearly in the manuscript on several points.

The work is comprehensive, well written, and meets the stated objectives. However, there are some minor issues to be improved:

- Figure and Table legends should be improved with more detailed descriptions because it is hard to understand what type of data are shown. For example, in Table 1: The title of the table should be more precise, as it does not specify what type of MITF data is presented.

- Also in Table 1: Mean (±SD) data should be presented according to the headliner, but data is shown in different way, i.e. “10.44 (9.4-11.3)”.

- In the “2.7 Differential gene expression analysis” section there are several descriptions about the examined genes, i.e: CXCL16, TIM3, CNIH4, HCP5, and so on, in my opinion these descriptions shoud be in the “Discussion” section of the manuscript.

Author Response

The manuscript entitled “Microphthalmia-associated transcription factor: a differentiation marker in Uveal Melanoma” describes an inverse correlation between MITF IHC staining and pigmentation of uveal melanoma, which associated with monosomy of chromosome 3/ loss of BAP1 expression. MITF is a well-known and well-studied regulator of cutaneous melanoma, and only a few studies address it in uveal melanoma. The aim of this study is clear, the results are well written, however, there are many limitations of the study which are stated clearly in the manuscript on several points.

The work is comprehensive, well written, and meets the stated objectives. However, there are some minor issues to be improved:

- Figure and Table legends should be improved with more detailed descriptions because it is hard to understand what type of data are shown. For example, in Table 1: The title of the table should be more precise, as it does not specify what type of MITF data is presented.

Reply: We thank the reviewer for the suggestion which helps to improve our paper. We have now added more details in the table and figure legends and clarified how gene expression was measured and how the “low” and “high” groups were created.

- Also in Table 1: Mean (±SD) data should be presented according to the headliner, but data is shown in different way, i.e. “10.44 (9.4-11.3)”.

Reply: Thank you for noticing! The expression values reported in Table 1 are, indeed, not Mean (±SD) but median (range). We have now amended this in the revised version of the manuscript.

- In the “2.7 Differential gene expression analysis” section there are several descriptions about the examined genes, i.e: CXCL16, TIM3, CNIH4, HCP5, and so on, in my opinion these descriptions shoud be in the “Discussion” section of the manuscript.

Reply: We have shortened the “Differential gene expression analysis” section and moved these gene descriptions to the discussion section (lines 386-444, manuscript with track changes).

Reviewer 2 Report (New Reviewer)

The article provides a clear and detailed analysis of the relationship between MITF expression and several genetic and clinical parameters in UM. The use of tables and figures also help in interrupting the results.

 My questions: Please provide detail 

Any inverse relationship suggest a potential therapeutic target for UM treatment?

 What is the relationship between MITF expression levels and survival in patients with UM, inflammatory phenotype or genetic affect this relationship?

 What implications do these findings have for the understanding of the relationship between MITF and pigmentation in UM?

 What is the relationship between MITF expression and EMT and stem cell markers in the cohort of UM patients?

 What is the significance of the downregulated genes IRS2, AHCYL2, PPP1R3C, SNHG7, and CDV3 in UM with low MITF expression?

 What is the significance of the enriched pathways in UM with low MITF, particularly those related to immune processes and metabolic reprogramming, and how do they relate to MITF expression?

 Additionally, why were EMT factors not among the most differentially expressed genes in the analysis and why was the Hallmark EPITHELIAL_MESENCHYMAL_TRANSITION pathway not significant?

Author Response

The article provides a clear and detailed analysis of the relationship between MITF expression and several genetic and clinical parameters in UM. The use of tables and figures also help in interrupting the results. 

 My questions: Please provide detail 

Reply: We thank the reviewer for the thoughtful comments and will do our best to address each question, with the knowledge we have at the moment. The line numbers mentioned in the reply refer to the manuscript version with track changes.

Any inverse relationship suggest a potential therapeutic target for UM treatment?

Reply: In the field of uveal melanoma, we are always in search of new targets that allow us to stop or decrease the metastatic burden. MITF is an attractive target, but one should be very careful because of its many functions in different cells and different cell states. Our results show that tumours with low MITF have more malignant features than tumours with high MITF, but evidence from cutaneous melanoma shows that MITF-high cells, while being less invasive, are highly proliferative. Moreover, there are contrasting reports on the role of MITF in the responsiveness or resistance to existing therapies. We believe that more studies are needed to pinpoint the most efficient way to target the MITF pathway and we do not wish to speculate too much. We did not any speculation to the text.

What is the relationship between MITF expression levels and survival in patients with UM, inflammatory phenotype or genetic affect this relationship?

Reply: We believe that MITF is involved in tumour behaviour but not independently. In our cohort, as in the study by Mouriaux et al., we did not find a significant difference in survival between UM with low and high MITF. However, a small group of 10 cases with very high MITF expression had excellent prognosis, when compared to the rest of the cases. Our data also show that 9 of these 10 cases have disomy 3. MITF expression is higher in disomy 3 than in monosomy 3 tumours and shows an inverse correlation with inflammatory markers. We believe that all these factors have a role and influence each other, with monosomy 3 being the chief driver of a poor prognosis and a prominent inflammatory microenvironment. We cannot say at this stage if MITF directly influences the expression of inflammatory markers or if the reverse is true. However, evidence from studies on cutaneous melanoma shows an inverse relation between MITF and inflammatory markers and that modulation of MITF expression causes changes in the inflammatory response [1-3]. These observations suggest that the modulation of MITF expression in uveal melanoma cells may be part of the process of tumour progression and metastasis formation, and one of the pathogenic routes following a BAP1 mutation/Chrom 3 loss.

We have now extensively discussed these concepts in the discussion section (lines 459-498).

 What implications do these findings have for the understanding of the relationship between MITF and pigmentation in UM?

Reply: MITF has many functions and targets in melanoblasts, melanocytes and melanoma cells, only some of which are related to melanin synthesis and pigmentation. In a recent review, we explored the previous literature regarding the role of MITF in melanocytes, cutaneous melanoma and uveal melanoma and the complex relationship between MITF and cellular processes such as cell survival, proliferation and invasion [4]. We found a negative correlation between MITF expression (both mRNA and protein) and pigmentation (and pigment-related genes). This suggests that in UM, the non-pigment-related function of MITF may be more relevant than its pigment-related functions.

We added a section in the discussion regarding this. (lines 359-361)

 What is the relationship between MITF expression and EMT and stem cell markers in the cohort of UM patients?

Reply: Previous studies in cutaneous melanoma showed that melanoma cells with high MITF have high TWIST1 and ZEB1 and are more invasive and less proliferative, while melanoma cells with low MITF have high ZEB2 and SNAI2.and are more proliferative and less invasive. A study by Asnaghi in UM corroborated this model in UM as well, with high TWIST1 being related to a poor prognosis. We now show that MITF expression levels are indeed negatively correlated to TWIST1 and positively correlated to SNAI2 and ZEB2, thus further corroborating the similarity with cutaneous melanoma. We found a positive correlation between BAP1 and SNAI2 expression and a negative correlation between BAP1 and TWIST1 expression (data not shown in the manuscript). These findings prompted us to postulate that BAP1 inactivation may be driving this EMT shift, possibly through MITF. Since these are in silico observations, we will be curious to see if this hypothesis can be confirmed in functional tests. This will be future work. These concepts are extensively explained in the discussion section of the manuscript (lines 499-525).

 What is the significance of the downregulated genes IRS2, AHCYL2, PPP1R3C, SNHG7, and CDV3 in UM with low MITF expression?

Reply: We have now updated the manuscript by moving gene descriptions to the discussion section. We hope that the new part in the discussion regarding clarifies their potential significance. IRS2 and PPP1R3C are involved in glycogen synthesis. Since GSEA showed that the metabolic pathways GYCOLYSIS, OXIDATIVE_PHOSPHORYLATION and KRAS_SIGNALING_UP and MTORC1_SIGNALING were enriched in MITF-low UM, we think tumours with low MITF may have a different metabolic landscape when compared to MITF-high UM. AHCYL2 and SNHG7 have been linked to proliferation arrest and cell cycle arrest (SNGH7 in UM cell lines specifically). Therefore, the downregulation of these two genes may be needed for the malignant progression of UM. A more detailed discussion of this topic can be found in the discussion section of the manuscript (lines 410-444).

 What is the significance of the enriched pathways in UM with low MITF, particularly those related to immune processes and metabolic reprogramming, and how do they relate to MITF expression?

 Reply: As discussed in the answer to the second question, previous studies have shown that MITF and the inflammatory response are related in cutaneous melanoma, with inflammatory markers influencing pigmentation and MITF expression and with modulation of MITF expression causing changes in the inflammatory response [1-3]. The fact that the most enriched pathways in MITF-low UM are related to immune processes lends support to these studies. A more prominent inflammatory infiltrate has been shown to be related to a worse prognosis in UM. We do not know if changes in MITF expression directly influence inflammatory markers in UM or if both phenomena occur as a consequence of a common upstream process related to loss of BAP1/chromosome 3. Nevertheless, we can hypothesise that, although not directly related to patient survival, MITF has a role in UM behaviour and that the modulation of its expression in UM cells may be part of the process of tumour progression and metastasis formation. (lines 459-498).

As for the pathways involved in metabolic reprogramming, the role of MITF in metabolism has not been extensively studied in the literature. Some studies have shown an association between MITF and hypoxia and have postulated that hypoxia and nutrient starvation may cause a decrease in MITF and greater invasiveness through HIF1α [5-7]. However, the HYPOXIA pathway is not differentially enriched in MITF-low vs MITF-high UM in our cohort. Haq et al. reported that activation of BRAF/MAPK pathway in cutaneous melanoma cells decreases MITF, which in turn decreases PGC1α, which is a potent activator of oxidative phosphorylation [8]. If a similar mechanism, possibly initiated by BAP1 inactivation, is present in UM as well, it may partially explain the different behaviour of tumours with high and low MITF. This point is discussed in the discussion section, lines 429-444.

 Additionally, why were EMT factors not among the most differentially expressed genes in the analysis and why was the Hallmark EPITHELIAL_MESENCHYMAL_TRANSITION pathway not significant?

Reply: EMT is an extremely complex process, and different factors have different and sometimes contrasting roles in different types of cancers, as extensively explained in the review by Nieto et al (Nieto, M.A., Huang, R.Y., Jackson, R.A., Thiery, J.P., 2016. Emt: 2016. Cell 166, 21-45.). The work of Caramel, Deckener and Vandamme showed that in cutaneous melanoma, only a subset of EMT markers (ZEB1 and TWIST1) are related to invasion while other markers (ZEB2 and SNAI2) act as tumour suppressors and are positively related to MITF. Therefore, the EMT process is extremely heterogeneous and the Hallmark pathway EPITHELIAL_MESENCHYMAL_TRANSITION does not distinguish between specific subsets of EMT markers. This may be the reason why EPITHELIAL_MESENCHYMAL_TRANSITION is not significantly enriched in MITF-low tumours. As for the lack of EMT markers among the most differentially expressed genes, we need to take into account the strict cut-offs of differential gene expression analysis, which uses FDR and multiple testing correction. While it is useful to identify genes that show highly significant differences between the two groups, it may dismiss genes with lower significance that may be part of important processes.

We did not extend the discussion on this subject as it already has quite a long text about this, and it would be quite speculative.

  1. Arts, N.; Cane, S.; Hennequart, M.; Lamy, J.; Bommer, G.; Van den Eynde, B.; De Plaen, E. microRNA-155, induced by interleukin-1ss, represses the expression of microphthalmia-associated transcription factor (MITF-M) in melanoma cells. PLoS One 2015, 10, e0122517, doi:10.1371/journal.pone.0122517.
  2. Riesenberg, S.; Groetchen, A.; Siddaway, R.; Bald, T.; Reinhardt, J.; Smorra, D.; Kohlmeyer, J.; Renn, M.; Phung, B.; Aymans, P.; et al. MITF and c-Jun antagonism interconnects melanoma dedifferentiation with pro-inflammatory cytokine responsiveness and myeloid cell recruitment. Nat Commun 2015, 6, 8755, doi:10.1038/ncomms9755.
  3. Wiedemann, G.M.; Aithal, C.; Kraechan, A.; Heise, C.; Cadilha, B.L.; Zhang, J.; Duewell, P.; Ballotti, R.; Endres, S.; Bertolotto, C.; et al. Microphthalmia-Associated Transcription Factor (MITF) Regulates Immune Cell Migration into Melanoma. Transl Oncol 2019, 12, 350-360, doi:10.1016/j.tranon.2018.10.014.
  4. Gelmi, M.C.; Houtzagers, L.E.; Strub, T.; Krossa, I.; Jager, M.J. MITF in Normal Melanocytes, Cutaneous and Uveal Melanoma: A Delicate Balance. Int J Mol Sci 2022, 23, doi:10.3390/ijms23116001.
  5. Cheli, Y.; Giuliano, S.; Fenouille, N.; Allegra, M.; Hofman, V.; Hofman, P.; Bahadoran, P.; Lacour, J.P.; Tartare-Deckert, S.; Bertolotto, C.; et al. Hypoxia and MITF control metastatic behaviour in mouse and human melanoma cells. Oncogene 2012, 31, 2461-2470, doi:10.1038/onc.2011.425.
  6. Cheli, Y.; Guiliano, S.; Botton, T.; Rocchi, S.; Hofman, V.; Hofman, P.; Bahadoran, P.; Bertolotto, C.; Ballotti, R. Mitf is the key molecular switch between mouse or human melanoma initiating cells and their differentiated progeny. Oncogene 2011, 30, 2307-2318, doi:10.1038/onc.2010.598.
  7. Feige, E.; Yokoyama, S.; Levy, C.; Khaled, M.; Igras, V.; Lin Richard, J.; Lee, S.; Widlund Hans, R.; Granter Scott, R.; Kung Andrew, L.; et al. Hypoxia-induced transcriptional repression of the melanoma-associated oncogene MITF. Proceedings of the National Academy of Sciences 2011, 108, E924-E933, doi:10.1073/pnas.1106351108.
  8. Haq, R.; Shoag, J.; Andreu-Perez, P.; Yokoyama, S.; Edelman, H.; Rowe, G.C.; Frederick, D.T.; Hurley, A.D.; Nellore, A.; Kung, A.L.; et al. Oncogenic BRAF regulates oxidative metabolism via PGC1α and MITF. Cancer Cell 2013, 23, 302-315, doi:10.1016/j.ccr.2013.02.003.

This manuscript is a resubmission of an earlier submission. The following is a list of the peer review reports and author responses from that submission.

Round 1

Reviewer 1 Report

In this manuscript, Gelmi et al. studied the relation between MITF and uveal melanoma development. MITF has been first described as the major regulator of cutaneous melanocyte differentiation. Then, this transcription factor has been involved in melanoma, regulating proliferation, invasion, inflammation cytokines and stemness phenotype. In this study, authors correlated the loss of MITF expression with the increase of inflammatory, EMT genes and survival, by analyzing a cohort of 64 patients by histology and genes expression studies. They concluded that uveal melanoma behave like cutaneous melanoma, meaning that loss of MITF induce dedifferentiation and increase EMT profile with inflammation.

The observations are really of interest for the fields, however some observations need to be strengthened as the observations are only correlative.

The regulation of the genes shown to be modulated by MITF (pigmentation, EMT and inflammatory) in the article need to be confirmed by modulating expression of MITF in cells models. Without these confirmations, the conclusions are falling short as it is only correlative.

For inflammatory genes, authors focused only on genes expression of immune cells (cells type ad some genes regulating activation of immune cells). MITF has been shown in cutaneous to regulate inflammatory cytokines such as CCL2, CYR61… Authors should focus on inflammatory cytokines shown to be regulated by MITF to conclude that uveal MITF regulate also inflammation as in cutaneous melanoma.

Author Response

In this manuscript, Gelmi et al. studied the relation between MITF and uveal melanoma development. MITF has been first described as the major regulator of cutaneous melanocyte differentiation. Then, this transcription factor has been involved in melanoma, regulating proliferation, invasion, inflammation cytokines and stemness phenotype. In this study, authors correlated the loss of MITF expression with the increase of inflammatory, EMT genes and survival, by analyzing a cohort of 64 patients by histology and genes expression studies. They concluded that uveal melanoma behave like cutaneous melanoma, meaning that loss of MITF induce dedifferentiation and increase EMT profile with inflammation.

The observations are really of interest for the fields, however some observations need to be strengthened as the observations are only correlative.

Answer: We thank the reviewer for the comments, which helped us to improve our paper. We rephrased some parts to highlight that our observations are correlations and not proof of causation.

  • Lines 393-395: Since these observations are based on in silico correlations, functional studies are needed to confirm if modulation of MITF expression directly leads to a change in the expression of inflammatory markers
  • Lines 428-430: As stated above, these in silico results should be confirmed by functional studies, possibly through modulation of MITF expression

The regulation of the genes shown to be modulated by MITF (pigmentation, EMT and inflammatory) in the article need to be confirmed by modulating expression of MITF in cells models. Without these confirmations, the conclusions are falling short as it is only correlative.

Answer: The reviewer points out an important consideration. Our results are indeed based on the in silico analysis of pre-collected data. In order to state that MITF directly influences inflammation and EMT, functional studies would be most welcome. As we hope that our findings may stimulate further research, we added two comments in the discussion section:

  • Lines 393-395: Since these observations are based on in silico correlations, functional studies are needed to confirm if modulation of MITF expression directly leads to a change in the expression of inflammatory markers
  • Lines 428-430: As stated above, these in silico results should be confirmed by functional studies, possibly through modulation of MITF expression.

For inflammatory genes, authors focused only on genes expression of immune cells (cells type ad some genes regulating activation of immune cells). MITF has been shown in cutaneous to regulate inflammatory cytokines such as CCL2, CYR61… Authors should focus on inflammatory cytokines shown to be regulated by MITF to conclude that uveal MITF regulate also inflammation as in cutaneous melanoma.

Answer: We looked at prior papers to identify important cytokines in Uveal Melanoma studies:

  • Bronkhorst 2012 (doi: 10.1159/000334576): CCL2, CCL17, CCL22
  • Robertson 2017 (doi: 10.1016/j.ccell.2017.07.003): IFNg, CCL2, CCL3, CCL4, CXCL9, CXCL10, CXCL13, IL6, IL10 (also used by Gezgin 2022, doi: 10.1016/j.xops.2022.100132)

We also considered papers that studied the association between MITF and cytokines and chemokines, such as Wiederman 2019 (10.1016/j.tranon.2018.10.014) and Riesenberg 2015 (doi: 10.1038/ncomms9755).

We made a list of the following and looked at their expression levels: CCL2, CCL3, CCL4, CCL5, CXCL9, CXCL10, CXCL13, IL1B, IL6, IL10, TNFa.

A high enough expression for a comparison with MITF was observed for: CCL2, CCL5, CXCL10, IL10.

CCL2

CCL3

CCL5 pr1

CCL5 pr2

CXCL9

CXCL10

IL10

MITF

Coeff.

-0.119

-0.244

-0.371

-0.328

-0.064

-0.267

-0.105

Sign

0.35

0.052

0.003

0.008

00616

0.033

0.41

Although some of the cytokines we tested show a negative correlation that is in line with our other findings, we must keep in mind that these values may also come from immune cells present in the tumours. Therefore, we think that these values are not sufficient to make claims or statements. In order to get some biologically-relevant information, much more in-depth analyses should be performed, that would be better suited for a separate study.

In our study, we did not only correlate the expression of MITF and immune cell markers, but we also performed differential gene expression analysis and gene set enrichment analysis (GSEA) comparing MITF-low and MITF-high tumours. Our GSEA showed that that the most enriched pathways in MITF-low tumours are indeed inflammatory pathways (see figure 6). We believe that this finding strengthens our hypothesis that tumours with low MITF usually have higher inflammation that tumours with high MITF.

Reviewer 2 Report

The manuscript "Microphthalmia-associated transcription factor: a differentiation marker in Uveal Melanoma" is well written and documented.

Few comments/improvement are needed:

1. Too much comparison with cutaneous melanoma

2. The study is based on enucleated eyes, smaller tumors might have different results. A comment on this is necessary

3. There are no comments and references on gene expression profile

Author Response

The manuscript "Microphthalmia-associated transcription factor: a differentiation marker in Uveal Melanoma" is well written and documented.

Few comments/improvement are needed:

  1. Too much comparison with cutaneous melanoma

Answer: Thank you for your comment. Since MITF has mostly been studies in skin and in cutaneous melanoma, we believe it is sensible to use cutaneous melanoma as a term of comparison or a starting point for our analyses. We do agree, however, that cutaneous and uveal melanoma are two very different conditions and that they are driven by different pathways. We have shortened the text and rephrased several paragraphs of the discussion (lines 406-440).

  1. The study is based on enucleated eyes, smaller tumors might have different results. A comment on this is necessary

Answer: A comment has been added to the Limitations section (4.4, lines 541-543):

“Moreover, our analysis focused on enucleated tumours, which are usually larger than UM treated conservatively. Therefore, our results may not be completely reproducible in a cohort with smaller tumours.”

  1. There are no comments and references on gene expression profile

Answer: Thank you for the comment. We have added a reference to GEP in lines 40-41. The text now reads as follows:

“Moreover, it is possible to accurately stratify patients in two metastatic risk categories (Class 1 and Class 2) with a 15-gene expression profile [11,12].”

Reviewer 3 Report

The manuscript presented by Gelmi et al, entitled: “Microphthalmia-associated transcription factor: a differentiation marker in uveal melanoma” describes the importance of MITF as a marker associated with differentiation and survival in uveal melanoma.

The Authors analyzed clinical and histopathological data of 64 UM patients and reported several findings, including: low MITF expression was associated with dark tumor pigmentation; MITF expression was significantly lower in monosomy 3; patients with high MITF had a significantly better survival rates; finally, they found an association between inflammation genes and a decreased expression of MITF.

 The article is well written and effectively defines MITF as an important marker in UM.

Author Response

The manuscript presented by Gelmi et al, entitled: “Microphthalmia-associated transcription factor: a differentiation marker in uveal melanoma” describes the importance of MITF as a marker associated with differentiation and survival in uveal melanoma.

The Authors analyzed clinical and histopathological data of 64 UM patients and reported several findings, including: low MITF expression was associated with dark tumor pigmentation; MITF expression was significantly lower in monosomy 3; patients with high MITF had a significantly better survival rates; finally, they found an association between inflammation genes and a decreased expression of MITF.

 The article is well written and effectively defines MITF as an important marker in UM.

Answer: We thank the reviewer for taking the time to read our work and for appreciating the value of this study.

Round 2

Reviewer 1 Report

In the first review, the main critical point was that the study is mainly in silico and correlative. In vitro experiments were suggested to confirm and straighten the observations. Authors didn’t perform the requested experiments and rephrased some paragraphs of the paper to not over interpret their conclusions. Without these in vitro experiments, the conclusions are still falling short and not sufficient for publication in this present form.

Author Response

In the first review, the main critical point was that the study is mainly in silico and correlative. In vitro experiments were suggested to confirm and straighten the observations. Authors didn’t perform the requested experiments and rephrased some paragraphs of the paper to not over interpret their conclusions. Without these in vitro experiments, the conclusions are still falling short and not sufficient for publication in this present form.

Reply: The reviewer raises an important issue. However, the aim of the present study was to perform an exploratory analysis of clinical data on an interesting molecule (MITF) using a cohort of patients with very long follow up. Our analysis identified potentially interesting pathways which can be further tested and hopefully targeted. Since our results are mainly in silico and exploratory, we are planning on performing in vitro experiments for functional confirmation. This next phase however requires extensive testing with multiple techniques, both for modulation of MITF expression and for measurement of expression of potential target genes and proteins. For these reasons, we believe that these functional experiments would be better suited for a separate study.

Round 3

Reviewer 1 Report

 -